# Artesunate Inhibits Metastatic Potential in Cisplatin-Resistant Bladder Cancer Cells by Altering Integrins

**DOI:** 10.3390/cells14080570

**Published:** 2025-04-10

**Authors:** Olesya Vakhrusheva, Fuguang Zhao, Sascha Dennis Markowitsch, Kimberly Sue Slade, Maximilian Peter Brandt, Igor Tsaur, Jindrich Cinatl, Martin Michaelis, Thomas Efferth, Roman Alexander Blaheta, Axel Haferkamp, Eva Juengel

**Affiliations:** 1Department of Urology and Pediatric Urology, University Medical Center Mainz, Langenbeckstr. 1, 55131 Mainz, Germany; olesya.vakhrusheva@med.uni-tuebingen.de (O.V.); zhaofuguang1990@gmail.com (F.Z.); sascha.markowitsch@unimedizin-mainz.de (S.D.M.); kimberlysue.slade@unimedizin-mainz.de (K.S.S.); maximilian.brandt@unimedizin-mainz.de (M.P.B.); igor.tsaur@med.uni-tuebingen.de (I.T.); roman.blaheta@unimedizin-mainz.de (R.A.B.); axel.haferkamp@unimedizin-mainz.de (A.H.); 2Department of Urology, University Hospital Tübingen, Hoppe-Seyler-Strasse 3, 72076 Tübingen, Germany; 3Interdisciplinary Laboratory for Paediatric Tumour and Virus Research, Dr. Petra Joh Research Institute, 60529 Frankfurt am Main, Germany; j.cinatl@kinderkrebsstiftung-frankfurt.de (J.C.J.); m.michelis@kent.ac.uk (M.M.); 4School of Natural Sciences, University of Kent, Canterbury CT2 7NJ, UK; 5Institute of Pharmaceutical and Biomedical Sciences, Johannes Gutenberg University Mainz, Staudinger Weg 5, 55128 Mainz, Germany; efferth@uni-mainz.de

**Keywords:** bladder cancer (BCa), artesunate (ART), cisplatin resistance, adhesion, chemotaxis, migration, invasion, integrin

## Abstract

The survival of patients with locally advanced and metastatic bladder cancer (BCa) is persistently low. Hence, new treatment options are urgently needed. Artesunate (ART) a derivative of artemisinin, used in Traditional Chinese Medicine, shows anti-tumor activity extending over a broad spectrum of human cancers. As we have previously shown, ART inhibits growth in cisplatin-sensitive (parental) and cisplatin-resistant BCa cells. However, how ART acts on the metastatic potential of BCa remained unclear. To clarify, we applied ART to parental and cisplatin-resistant RT4, RT112, T24, and TCCSup BCa cell lines. We examined tumor cell adhesion to vascular endothelium and immobilized collagen and evaluated chemotactic activity, migration, and invasive activity of the BCa cells. Adhesion receptors, integrin α and β subtypes, integrin-linked kinase (ILK), and focal adhesion kinase (FAK) were investigated. The functional relevance of integrin expression altered by ART was determined by blocking studies. ART significantly reduced tumor cell adhesion to vascular endothelium and immobilized collagen in parental as well as in cisplatin-resistant BCa cells. Depending on cell type, ART suppressed tumor cell motility and diminished integrin expression (surface and total). Functional blocking of integrins altered by ART reduced cell adhesion and invasion of the BCa cells. Thus, the metastatic potential of parental and cisplatin-resistant BCa cells was significantly inhibited by ART, making it a promising treatment option for patients with advanced or therapy-resistant BCa.

## 1. Introduction

Bladder cancer (BCa), accounts for 3% of global cancer diagnoses, with the highest incidence rates in south and west Europe as well as in North America [1]. Close to 5% of BCa patients have metastatic disease at the time of diagnosis [2], with a much higher proportion reported in western countries [3]. Additionally, approximately half of patients with muscle-invasive bladder cancer (MIBC) relapse with distant metastases after radical cystectomy. This disease stage is characterized by very poor prognosis [4]. The average 5-year survival rate for BCa in western Europe is ~60%, while, for metastatic BCa, it is less than 10% [5]. Cisplatin-based chemotherapy is the first-line treatment for eligible patients with locally advanced and metastatic BCa, with an initial response rate of ~50% [6,7]. However, due to the emergence of cisplatin resistance, disease recurrence and metastasis often occur, leading to treatment failure. Thus, preventing and overcoming cisplatin resistance could provide a distinct advantage in regard to overall life quality and survival for patients with advanced, muscle-invasive BCa. Hence, a new approach for patients with advanced and cisplatin-resistant BCa is important.

Complementary and alternative medicine, as an effective option, is becoming increasingly popular for cancer therapy worldwide [8,9]. In Europe, approximately 40% of cancer patients apply complementary and alternative medicine, primarily to reduce adverse side effects, enhance the immune system, and increase the effectiveness of conventional therapy [10,11]. Natural compounds and their derivatives, as a type of complementary and alternative medicine, provide rich resources of potential adjuvant cancer therapy. In preliminary studies, natural compounds and their derivatives have enhanced the anti-tumor effects of established therapy or even counteracted therapy resistance [12,13,14,15,16,17].

Artemisinin, a natural compound extracted from the wormwood plant, an effective anti-malarial drug, has been shown to exhibit selective anti-tumor properties [18]. Artesunate (ART), a semi-synthetic derivative of artemisinin, has better biopharmaceutical properties, bioavailability, fewer side effects, and is well-tolerated and widely used for anti-malarial therapy [19,20]. ART has been shown to exert anti-tumor activity towards a broad spectrum of human cancers [18], including BCa [21]. The sensitivity of different tumors to conventional therapies has also been enhanced by ART [16,17,22,23]. Previous studies [23,24,25,26] report that ART induced apoptosis or ferroptosis in different therapy-resistant tumors. In line with these findings, we previously found that ART impairs growth of cisplatin-sensitive and cisplatin-resistant BCa cells through cell cycle arrest and apoptosis induction [21]. Moreover, ART has displayed remarkable anti-metastatic properties in various tumor cell lines. It has reduced migration in esophageal squamous cell carcinoma [27] and inhibited cell growth, clonogenicity, invasion, and angiogenesis in metastatic renal cell carcinoma in vitro and in vivo [28]. In prostate cancer cells, ART has induced cell apoptosis and suppressed cell migration by down-regulating urothelial carcinoma-associated (UCA1) lncRNA [29]. ART has been shown to inhibit lung tumorigenesis and tumor metastasis through Wnt/β-catenin signaling [30]. Moreover, combined with bromocriptine, ART exerted a synergistic effect in pituitary adenoma cells where cell migration and invasion were attenuated by suppressing miR-200c and stimulating PTEN expression [17].

Still, the anti-metastatic mechanism of ART in therapy-sensitive and therapy-resistant BCa cells remains unclear. Therefore, the present study was designed to investigate molecular alterations when ART blocks metastatic properties in a panel of parental and cisplatin-resistant BCa cell lines.

## 2. Materials and Methods

### 2.1. Cell Culture

Cisplatin-sensitive (parental) BCa cell lines RT4 (grade 1), RT112 (grade 2/3), T24 (grade 3), and TCCSup (grade 4) were purchased from DSMZ. Their cisplatin-resistant sublines were obtained from RCCL collection and were maintained as previously described [21,31,32]. RT4 and RT112 represent low-grade, well-differentiated bladder cancer cells with lower metastatic potential, while T24 and TCCSup are high-grade, poorly differentiated cell lines, exhibiting greater metastatic capabilities and more aggressive characteristics [33]. HUVECs (human umbilical vein endothelial cells) were isolated from umbilical cords provided by the Department of Gynecology at the University Medical Center Mainz, Germany. HUVECs were subcultured up to passage 5, as published [34].

### 2.2. Resistance Induction and Drug Treatment

Cisplatin-resistant sublines were continuously supplemented with 1 μg/mL cisplatin (Selleckchem, Munich, Germany) [21]. Thereby, cisplatin resistance differs in strength among the four BCa cell lines, as shown previously [35]. Parental and cisplatin-resistant cells were treated with ART (Sigma-Aldrich, Darmstadt, Germany) for 48 h or 72 h at a concentration of 2.5 or 10 μM, as previously published [21]. ART-untreated parental and cisplatin-resistant BCa cells served as controls.

### 2.3. Tumor Cell Adhesion to Collagen and HUVECs

The interaction of tumor cells with collagen and HUVECs was investigated as reported. Briefly, 24-well plates were overnight pre-coated with 200 μg/mL of collagen G (Merck, Darmstadt, Germany). Wells incubated with PBS were used as the background control. To prevent unspecific cell adhesion, the wells were incubated with 1% BSA (Sigma-Aldrich, Darmstadt, Germany) for 1 h. To detect the binding capacity of tumor cells to vascular endothelium, a total of 1.25 × 10^5^ HUVECs per well were seeded onto 24-well plates 16 h prior to the adhesion assay.

For the adhesion assay, BCa cells were pre-treated with ART or diluent for 48 or 72 h and labeled with Molecular Probes Cell Tracker Green CMFDA Dye (Thermo Fisher Scientific, Darmstadt, Germany). A total of 1.25 × 10^5^ tumor cells was added per well and incubated for 30 min (collagen G) or 2 h (HUVECs) at 37 °C in a humidified CO_2_ incubator. The wells were then washed with warmed (37 °C) PBS containing Ca^2+^ and Mg^2+^ (Sigma-Aldrich, Darmstadt, Germany) and fixed using 2% glutaraldehyde (Sigma-Aldrich, Darmstadt, Germany). A Sapphire Imager (Azure Biosystems, Munich, Germany) was used to determine the relative fluorescence intensity of attached cells, and the data were analyzed with the Azure Spot program (Azure Biosystems, Munich, Germany). The values were expressed in percent and normalized to untreated controls, set to 100%.

### 2.4. Tumor Cell Motility

Cell motility, including chemotaxis, migration, and invasion were evaluated using Falcon^®^ 24-well Companion plates and Corning^®^ FluoroBlok Inserts with 8 µm pore size (both: Corning GmbH, Kaiserslautern, Germany). A higher FCS medium concentration (30%) was placed in the lower chamber as a chemoattractant for all three cell motility assays. For chemotaxis, inserts remained uncoated. Inserts for migration were coated with 200 μg/mL of collagen G (Biochrom, Berlin, Germany) overnight at 4 °C and blocked with 1% BSA (Sigma-Aldrich, Darmstadt, Germany). For invasion, inserts were coated with 200 μg/mL of collagen G (Biochrom, Berlin, Germany) for 1 h at room temperature, then loaded with HUVECs and incubated overnight with HUVECs medium.

Parental and cisplatin-resistant BCa cells, pre-treated with ART or diluent, were stained with CellTracker Green CMFDA Dye for 30 min at 37 °C. A total of 6 × 10^4^ BCa cells were added per insert. After 24 h incubation, the inserts were washed with warmed (37 °C) PBS containing Ca^2+^ and Mg^2+^ (Sigma-Aldrich, Darmstadt, Germany) and fixed using 2% glutaraldehyde (Sigma-Aldrich, Darmstadt, Germany). The relative fluorescent intensity was measured using a Sapphire Imager (Azure Biosystems, Munich, Germany) to score cell motility. The data were analyzed with the Azure Spot program (Azure Biosystems, Munich, Germany), and presented in percent, normalized to corresponding untreated controls, set to 100%.

### 2.5. Integrin Surface Expression

Parental and cisplatin-resistant BCa cells, treated with ART or diluent for 48 h, were washed with FACS buffer (0.5% BSA in PBS) and then incubated with phycoerythrin (PE)-conjugated monoclonal antibodies for 1 h at 4 °C. The following antibodies directed against the integrin subtypes were used: anti-α1 (mouse IgG1; clone SR84), anti-α2 (mouse IgG2a; clone 12F1), anti-α3 (mouse IgG1; clone C3II.1), anti-α4 (mouse IgG1; clone 9F10), anti-α5 (mouse IgG1; clone IIA1), anti-α6 (rat IgG2a; clone GoH3; all: BD Biosciences, Heidelberg, Germany), anti-αv (rabbit IgG; clone EPR16800; Abcam, Berlin, Germany), anti-β1 (mouse IgG1; clone MAR4), anti-β3 (mouse IgG1; clone VI-PL2), anti-β4 (rat IgG2b; clone 439–9B; all: BD Biosciences, Heidelberg, Germany), and anti-β5 (mouse IgG2a; clone F-5; Santa Cruz Biotechnology, Heidelberg, Germany). The fluorescent intensity of 1 × 10^4^ cells was measured using a flow cytometer (Fortessa X20, BD Biosciences, Heidelberg, Germany) and expressed as mean MFI (mean fluorescence intensity). Mouse IgG1-PE (MOPC-21), IgG2a-PE (G155–178), rat IgG2a-PE (R35-95), rat IgG2b-PE (R35-38; all: BD Biosciences, Heidelberg, Germany), and rabbit IgG (Polyclonal; Abcam, Berlin, Germany) served as isotype controls.

### 2.6. Western Blot

Western blot analysis was used to investigate the total expression of integrin subtypes. Fifty µg of each tumor cell lysate was loaded onto 7% polyacrylamide gels, concentrated at 80 V for 10 min, separated at 120–150 V and transferred to polyvinylidenfluoride (PVDF) membranes (1 h, 100 V). The membranes were blocked with non-fat dry milk for 1 h and incubated overnight with monoclonal primary antibodies directed against integrins and integrin-related signaling proteins: anti-α3 (mouse IgG2a, dilution 1:1000, 150 kDa, Santa Cruz Biotechnology, Heidelberg, Germany), anti-α6 (rabbit, dilution 1:1000, 150/125 kDa, Cell signaling, Frankfurt am Main, Germany), anti-α2 (mouse IgG2a, clone 2, dilution 1:1000, 150 kDa), anti-β1 (mouse IgG1, clone 18, dilution 1:1000, 130 kDa), anti-specific focal adhesion kinase (FAK) (mouse IgG1, clone 77, dilution 1:1000, 116–125 kDa), anti-phospho-specific focal adhesion kinase (pFAK) (mouse IgG1, clone 14, dilution 1:1000, 116–125 kDa), and anti-integrin-linked kinase (ILK) (mouse IgG1, clone 3, dilution 1:1000, 50 kDa, all: BD Biosciences, Heidelberg, Germany). HRP-conjugated rabbit anti-mouse IgG or goat anti-rabbit IgG served as secondary antibodies (IgG, both: dilution 1:1000, Dako, Glosturp, Denmark). The membranes were incubated with ECL detection reagent (AC2204, Azure Biosystems, Munich, Germany) to visualize proteins with a Sapphire Imager (Azure Biosystems, Munich, Germany). AlphaView software (Version 3.4.0.0) (ProteinSimple, San Jose, CA, USA) was used to quantify the signals. Only images below a maximum band intensity of the device specific maximum were evaluated. The ratio of protein intensity/whole protein intensity (Coomassie stain of the membrane) was calculated and expressed in percent; corresponding untreated controls were set to 100%.

### 2.7. Integrin Signaling Blockade

To analyze the functional relevance of ART-altered integrins on the metastatic potential of the BCa cells, these integrins were blocked by function blocking antibodies on the tumor cells (without ART treatment), and adhesion and invasion assays were performed. In brief, parental and cisplatin-resistant RT112 and T24 cells were incubated for 1 h at 37 °C in a humidified CO_2_ incubator with 10 μg/mL of function-blocking anti–integrin α2 (clone P1E6), anti-integrin α3 (clone P1B5), anti-integrin α6 (clone NKI-GoH3), or anti-integrin β1 (clone 6S6; all: Sigma-Aldrich, Darmstadt, Germany) antibodies. Thereafter, tumor cell adhesion and invasion were analyzed as described above (see Section 2.3 and Section 2.4). Untreated cells served as controls.

### 2.8. Statistics

All experiments were performed at least three times. Evaluation, generation of mean values, and normalization were all carried out using Microsoft Excel. GraphPad Prism 7.0 (GraphPad Software Inc., San Diego, CA, USA) was used to investigate statistical significance (*p* ≤ 0.05) as follows: the two-way ANOVA test (adhesion to collagen and HUVECs, tumor cells chemotaxis, migration and invasion, Western blot and basal integrin expression) or two-sided *t*-test (integrin expression after ART treatment and integrin blockage studies).

## 3. Results

### 3.1. ART Significantly Inhibited Bladder Cancer Cell Adhesion

The current study investigated the effect of ART on BCa cell adhesion to collagen, an extracellular matrix protein, and HUVECs, representing vascular endothelial cells. Adhesion of both parental and cisplatin-resistant RT4, T24, and TCCSup cells to collagen was inhibited by ART in a dose-dependent manner (Figure 1a,c,d). Also, adhesion to collagen of cisplatin-resistant RT112 cells was reduced compared to controls, but independently from the ART concentration (Figure 1b). In contrast, parental RT112 cells were not affected by low or high ART dosage (Figure 1b). Accordingly, only the higher concentration of ART [10 µM] was chosen in further exploration.

Tumor cell adhesion to HUVECs was significantly reduced in all BCa cell lines after exposure to ART, compared to untreated controls (Figure 2). ART suppressed the adhesion ability of cisplatin-resistant RT112 and T24 cells to HUVECs more effectively than that of their parental counterparts (Figure 2b,d). Inhibition of the binding capacity of parental and cisplatin-resistant RT4 and TCCSup cells to HUVECs by ART was comparable (Figure 2a,c).

### 3.2. ART Impaired Chemotaxis, Migration, and Invasion of Bladder Cancer Cells

Tumor cell motility including chemotaxis, migration, and invasion was investigated in BCa cells after ART application. ART moderately, but significantly, inhibited chemotactic activity of parental RT4 and cisplatin-resistant RT112 cells, whereas no changes were detected in the other BCa cell lines, compared to untreated controls (Figure 3). Furthermore, significant suppression of cell migration was found in both parental and cisplatin-resistant RT4, RT112, and T24 cells after treatment with ART, compared to untreated controls (Figure 3a–e). ART also caused inhibition of the invasive behavior in parental and cisplatin-resistant RT4, RT112, and T24 cells (Figure 3a–e). Cell motility was not altered in parental or cisplatin-resistant TCCSup cells after treatment with ART (Figure 3b,f).

From the utilized cell lines, the metastatic potential of RT112 and T24 cells was most strongly inhibited by ART. Therefore, these two cell lines were examined in more detail.

### 3.3. ART Suppressed the Cell Surface Expression of Integrins

Since tumor cell adhesion and motility are regulated by integrin receptors α- and β-subtypes, the basal expression of integrins on the BCa cell surface was analyzed. Integrins α2, α3, α6, β1, and β4 were highly expressed in parental and cisplatin-resistant RT112 cells (Figure 4a,c). However, the expression of integrins α2, α3, and β1 was lower in cisplatin-resistant RT112 cells than in parental cells (Figure 4a,c). ART treatment caused a significant reduction of integrin α2, α3, α6, β1, and β4 expression on the cell surface of both parental and cisplatin-resistant RT112 cells (Figure 4e). Relatively low expression of integrin α1, α4, α5, αv, β3, and β5 was apparent on the cell surface of both parental and cisplatin-resistant RT112 cells and was therefore not calculated after ART application (Figure 4a,c).

In parental and cisplatin-resistant T24 cells, the integrins α2, α3, α6, and β1 were also highly expressed on the cell surface at steady-state (Figure 4b,d). Notably, the expression of integrins α2, α3, α6, and β1 was lower in the cisplatin-resistant T24 cells, compared to the parental T24 cells (Figure 4b,d). Integrin α5 was moderately expressed on the cell sur-face in both parental and cisplatin-resistant T24 cells compared to the other integrins. Pa-rental T24 cells, but not cisplatin-resistant T24, expressed integrin β4 at a low level (Figure 4a,c). No expression of integrins α1, α4, αv, β3, and β5 was detectable on the cell surface of parental and cisplatin-resistant cells (Figure 4a–d). Similar to RT112 cells, the levels of integrins α2, α3, α5, α6, and β1 were significantly down-regulated by ART in both parental and cisplatin-resistant T24 cells (Figure 4f). Expression of integrin β4 was significantly reduced by ART in parental T24 cells (Figure 4f).

**Figure 4 cells-14-00570-f004:**
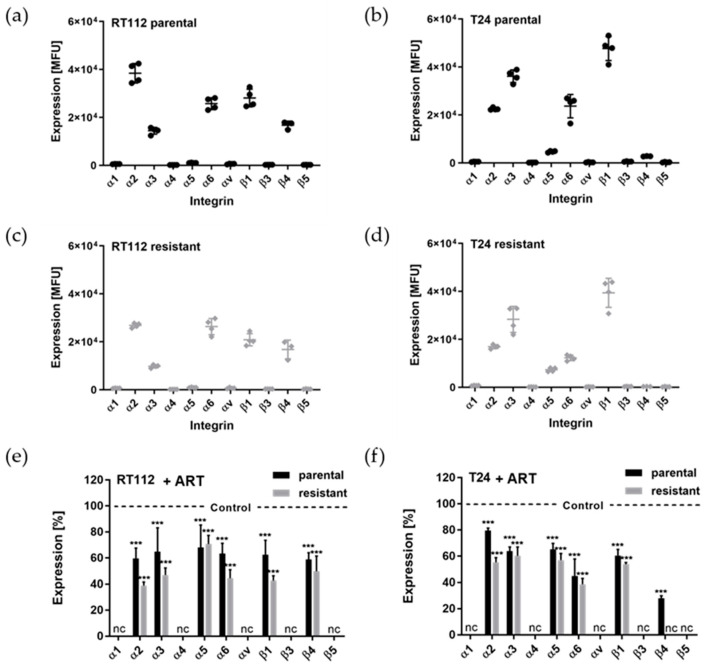
Surface expression of integrin subtypes in the BCa cells and its modification by ART. The basal surface expression of integrin α- and β-subtypes in untreated parental and cisplatin-resistant RT112 (**a**,**c**) and T24 (**b**,**d**) cells. Surface expression of integrin subtypes in RT112 (**e**) and T24 (**f**) cells after 48 h ART [10 µM] exposure, compared to untreated controls. MFUs (mean fluorescence units) of untreated controls were set to 100% (dotted line). Error bars indicate standard deviation (SD). Significant difference to untreated control: *** = *p* ≤ 0.001. nc = not calculated. *n* = 4.

Integrins α2, α3, α6, and β1 were highly expressed in both RT112 and T24 cells, and their expression significantly diminished after exposure to ART. Thus, the following investigation of the total integrin expression focused on these four integrin subtypes.

### 3.4. Modifications of the Total Integrin Content by ART

In accordance with the surface expression of the chosen integrin subtypes, ART significantly diminished the total content of integrin α2, α3, and β1 in RT112 cells. Overall, the ART-mediated down-regulation of integrin expression was more pronounced in cisplatin-resistant RT112 cells (Figure 5a,c). Of note, the level of integrin α6 was also down-regulated after treatment with ART in cisplatin-resistant RT112 cells, whereas it was elevated in parental cells (Figure 5a,c). The expression of the integrin-related signaling proteins FAK, pFAK, and ILK was down-regulated by ART in parental as well as in cisplatin-resistant RT112 cells (Figure 5a,c).

In T24 cells, ART significantly diminished the expression of total integrin α3 and α6 in both parental and cisplatin-resistant cells (Figure 5b,d). However, ART did not alter the expression of total integrin β1 in parental cells, whereas its expression moderately de-creased in the cisplatin-resistant cells after exposure to ART (Figure 5b,d). Integrin α2 expression remained unchanged after treatment with ART in both parental and cisplatin-resistant cells, compared to the controls (Figure 5b,d). The expression of the integrin-related signaling protein FAK slightly decreased after exposure to ART in the parental cells. pFAK did not alter upon ART application in the parental cells, whereas FAK and pFAK were up-regulated in cisplatin-resistant T24 cells (Figure 5b,d). In addition, ART slightly reduced the expression of ILK in both parental and cisplatin-resistant T24 cells (Figure 5b,d).

**Figure 5 cells-14-00570-f005:**
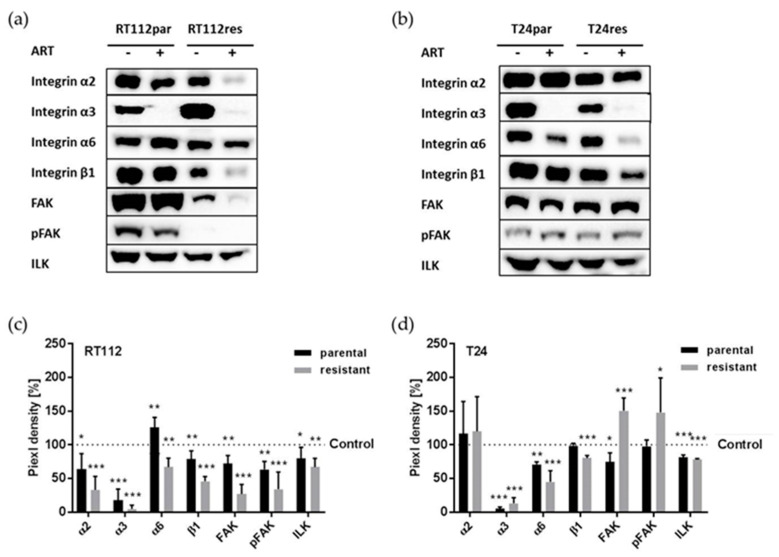
Total integrin expression of RT112 and T24 cells after ART treatment. Representative Western blot pictures of the integrin expression in parental (par) and cisplatin-resistant (res) RT112 (**a**) and T24 (**b**) cells after 48 h exposure to ART [10 µM]. Pixel density analysis of protein expression in RT112 (**c**) and T24 (**d**) cells, compared to the untreated controls (set to 100%, dotted line). Each protein analysis was accompanied by normalization to a total protein content of the cells. Error bars indicate standard deviation (SD). Significant difference to untreated control: * = *p* ≤ 0.05, ** = *p* ≤ 0.01, *** = *p* ≤ 0.001. *n* = 4. For detailed information, see Appendix A.

### 3.5. Integrin Blockade

Since ART caused inhibition of integrin expression in parental and cisplatin-resistant cells, we analyzed the impact of the relevant integrins on cell adhesion and invasion. Signaling of integrin α2, α3, α6, or β1 was blocked by using function-blocking antibodies in parental and cisplatin-resistant RT112 and T24 cells to evaluate their functional relevance. Cell adhesion to collagen and cell invasion through collagen and HUVECs were investigated (Figure 6). Blocking of integrin α2 signaling significantly reduced adhesion of RT112 and T24 cells, excepting parental RT112 cells (Figure 6a,b). In contrast, the invasive behavior of parental and cisplatin-resistant RT112 and T24 cells was not altered after blockade of integrin α2 (Figure 6c,d). Blocking integrin α3 in cisplatin-resistant RT112 and T24 cells led to inhibited invasive capacity, while no changes were detected in cell adhesion (Figure 6). Furthermore, disruption of integrin α6 signaling caused a reduction in cisplatin-resistant RT112 and T24 cells binding to collagen but did not in parental RT112 and T24 cells (Figure 6a,b). Parental RT112 cell invasive capacity was only slightly inhibited after blocking integrin α6 (Figure 6c,d). Both cell adhesion and invasive behavior decreased after incubation with function-interfering integrin β1 antibodies in cisplatin-resistant RT112 and T24 cells (Figure 6). Disruption of integrin β1 signaling resulted in extenuated adhesion to collagen of parental T24 cells (Figure 6b).

## 4. Discussion

Most cancer deaths are caused by metastatic disease. Cisplatin resistance plays a significant role in the dismal prognosis and progression of muscle-invasive bladder cancer. Approximately 60% of patients still exhibit muscle-invasive disease at cystectomy and, despite neoadjuvant chemotherapy, long-term survival remains poor [36]. Using cisplatin-resistant BCa cells as a model of cisplatin resistance in patients with advanced BCa, it is possible to evaluate new treatment options in vitro. ART has emerged as a promising therapeutic for BCa, demonstrating its ability to impair growth in cisplatin-resistant BCa cells through mechanisms including cell cycle arrest, apoptosis induction, and autophagy, potentially offering a novel approach to treat advanced and therapy-resistant bladder cancer [21]. In the current study, the inhibitory activity of ART on the metastatic potential of a panel of parental and cisplatin-resistant BCa cell lines was investigated.

Tumor cell interaction with extracellular matrix and vascular endothelium are important steps in cancer metastasis. Hence, interrupting these steps might prevent tumor cell dissemination. In the present study, ART significantly inhibited adhesion to collagen and HUVECs in parental and cisplatin-resistant BCa cells. In line with our findings, ART has been shown to inhibit attachment of melanoma cells [37] and liver cancer cells [38] to collagen. Artemisinin and ART have also been shown to block cell adhesion in other human diseases. In malaria patients, Souza and colleagues have shown that treatment with ART prevents the adherence of parasitized red blood cells to vascular endothelial cells [39].

Since tumor cell motility contributes to metastasis, attenuating the capacity of tumor cell motility might help to prevent metastasis. In the present study, we confirmed that ART mitigated the motility of BCa cells by inhibiting chemotaxis, migration, and invasion in a cell type-dependent manner. ART blocked moderately, but significantly, chemotaxis in parental RT4 and cisplatin-resistant RT112 cells. In line with our findings, a study with dihydroartemisinin, another semi-synthetic derivative of artemisinin, showed significantly different chemokine production that affected cell chemotaxis in a cerebral malaria murine model [40]. The capability of cells to migrate through collagen was significantly attenuated by ART in parental and cisplatin-resistant T24, RT4, and RT112 cells. In good accordance with our results, ART suppressed the migratory activity of lung [30], renal [28], prostate [29], and esophageal cancer cells [27] as well as of glioma [41] and pituitary adenoma cells [17]. Furthermore, in the current study, the invasive activity (through collagen and HUVECs) of parental and cisplatin-resistant RT4, RT112, and T24 cells was significantly diminished after exposure to ART. Consistently, ART has also been shown to suppress the invasive behavior of glioma [41], pituitary [17], and lung [30] cancer cells, as well as hemangioendothelioma cells [42]. In the current study, only collagen was employed as the extracellular matrix (ECM) to evaluate the migration and invasion of the BCa cells. Collagen is the major component of the ECM and tumor microenvironment [43,44,45,46] and is mainly involved in the migration and invasion [43,47] process. However, other ECM components are also involved in tumor metastasis. Thus, future studies with models including other ECM components might give further insight into the complex actions occurring during the metastatic process.

Integrin alterations were analyzed, since integrins play a key role in cell adhesion, migration, and invasion during tumor dissemination. In the current investigation, the basal surface expression of integrins was found to be cell type-dependent. Integrin α2, α3, α6, β1, and β4 were highly expressed in parental and cisplatin-resistant RT112 cells. T24 cells also highly expressed integrin α2, α3, α6, and β1. Furthermore, T24 cells revealed a moderate expression of integrin α5 on the cell surface. This observation is in line with previously reported cell type-dependent integrin expression in BCa cell lines [14,48]. ART significantly down-regulated cell surface expression of integrin α2 in parental and, to a greater extent, in cisplatin-resistant RT112 and T24 cells. ART altered not only the expression of integrin α2 at the cell surface but also the total protein expression in parental and cisplatin-resistant RT112 cells. In T24 cells, the total integrin α2 expression was not altered after ART exposure, indicating its internalization into the cell after exposure to ART.

The functional relevance of the ART-altered integrin surface expression on cell adhesion and invasion was demonstrated by blocking integrin α2. Suppression of integrin α2 reduced cell adhesion in cisplatin-resistant RT112 cells as well as in parental and cisplatin-resistant T24 cells. This finding is consistent with in vitro studies in head and neck [49] and prostate [50] cancer cells. Thus, ART-mediated inhibition of integrin α2 might account for the attenuated binding capacity of RT112 and T24 cells to collagen, except for parental RT112 cells that did not show this attenuation.

ART induced significant down-regulation not only of cell surface expression but also of total integrin α3 expression. In cisplatin-resistant RT112 and T24 cells, ART inhibited integrin α3 cell surface expression most strongly. Significantly inhibited invasive behavior by blocking integrin α3 with function-blocking antibodies was more pronounced in cisplatin-resistant RT112 and T24 cells than in parental cells. Similarly, shRNA-mediated knockdown of integrin α3 has been shown to reduce colorectal cancer cell invasion [51]. In glioma stem-like cells, integrin α3 was overexpressed and promoted cell invasion [52]. Integrin α3 has been associated with a more aggressive phenotype, a higher risk of recurrence, and poor prognosis in patients with colorectal cancer [53,54]. Thus, down-regulation of integrin α3 by ART might at least partially be responsible for less invasive behavior, especially of cisplatin-resistant RT112 and T24 cells.

ART caused significant reduction of integrin α6 expression on the cell surface in both parental and cisplatin-resistant RT112 and T24 cells. Also, the intracellular integrin α6 level was down-regulated by ART in parental and cisplatin-resistant T24 cells. In RT112 cells, integrin α6 was differentially regulated by ART, suggesting a different role in parental and cisplatin-resistant cells. In parental RT112 cells, the total integrin α6 expression increased after ART exposure. In contrast, cell surface integrin α6 decreased, indicating that ART induced translocation of integrin α6 from the cell membrane surface to the cytoplasm. In cisplatin-resistant RT112 cells, ART reduced both total and cell surface expression of integrin α6. Blockage of integrin α6 signaling resulted in a significant down-regulation of the adhesive capacity in cisplatin-resistant RT112 and T24 cells. In contrast, no changes were found in parental RT112 or T24 cells. In good accordance with these findings, in hepatocellular carcinoma cells, down-regulation of integrin α6 by shRNA interference decreased cell adhesion and invasion [55]. Moreover, in hemangioma stem cells, integrin α6 was required for adhesion to laminin and for vessel formation [56]. Thus, the ART-induced inhibition of tumor cell adhesion might at least partially be due to ART-reduced surface expression of integrin α6.

In the current investigation, the cell surface expression of integrin β1 was significantly down-regulated after ART application in parental and cisplatin-resistant RT112 and T24 cells. In contrast, alterations of intracellular integrin β1 levels were only found in cisplatin-resistant cells, again indicating translocation of integrin β1 away from the cell membrane surface. Integrin β1 blockage inhibited adhesion and invasion of cisplatin-resistant RT112 and T24 cells and diminished adhesion in parental T24 cells. Consistent with our data, cell adhesion and chemotaxis of four cisplatin-resistant BCa cell lines (TCCSup, HT1376, T24, and 5637) were down-regulated after blocking integrin β1 [57]. Previously, it has been shown that migration and invasion of T24 cells were reduced by α-lipoic acid via inhibition of integrin β1 cell surface expression [58]. Furthermore, blocking integrin β1 signaling has been shown to attenuate the binding and invasive capacity of 5637 BCa cells [59]. The BCa grade and worse clinical outcomes of BCa patients has been positively correlated with a high expression level of integrin β1 [60]. Thus, loss of integrin β1 signaling may be responsible for the decreased binding capacity and invasive behavior caused by ART in cisplatin-resistant RT112 and T24 cells.

Notably, in parental and cisplatin-resistant RT112 cells, the expression of mediators of the intracellular integrin signaling, FAK, pFAK, and ILK, was down-regulated after ART application. Moreover, ART decreased FAK in parental T24 cells and ILK in parental and cisplatin-resistant T24 cells. A previous study has shown that over-expression of FAK indicates a more aggressive behavior of BCa [61]. Thus, ART-induced depletion of FAK might induce reversion of cisplatin-resistant BCa cells to a less aggressive/normal cell phenotype. Another investigation has reported that ART’s down-regulation of ILK expression was correlated with inhibition of metastasis in BCa cells [61]. However, the expression of FAK and pFAK in cisplatin-resistant T24 cells was up-regulated after exposure to ART. Since FAK mediates diverse signaling pathways, this might represent another cell type-specific mechanism by which ART blocks adhesion. After ART treatment, reduced adhesion, migration, and invasiveness of cisplatin-resistant T24 cells seems to corroborate this hypothesis.

To summarize, ART suppressed RT112 and T24 cell adhesion by inhibiting the surface integrins α2, α6, and β1. RT112 and T24 cell invasion diminished after exposure to ART through down-regulation of integrin α3 and β1 surface expression. The parental RT112 cell adhesion was not altered by blocking the integrins, which is consistent with the results of ART-induced cell adhesion. ART suppressed the invasive behavior of parental RT112 and T24 cells, whereas blocking integrins did not alter cell invasiveness. Thus, ART-altered cell invasion in parental RT112 and T24 cells might be regulated by several kinds of integrins and/or integrin endocytic trafficking. Notably, blocking of integrin β1 exhibited significantly stronger anti-metastatic activity in RT112 and T24 cells, indicating that ART inhibits metastatic processes in BCa cells, mainly by modulating integrin β1-related signaling pathways. Integrins and integrin signaling proteins like FAK and ILK have also been shown to be involved in metastatic processes in BCa in vivo. Likewise, targeting integrin α6 significantly inhibited BCa cell migration in vivo [62]. Anti-tumor treatment of BCa with homoharringtonin reduced cell-extracellular matrix interaction, cell migration, and metastatic progression and was associated with inactivation of the integrin-FAK axis in vitro and in vivo [63]. Moreover, integrin-linked kinase (ILK) has been shown to be up-regulated in BCa cells, promoting epithelial–mesenchymal transition (EMT) in vitro and in vivo, leading to enhanced metastasis [64].

Overall, ART has demonstrated efficacy across a panel of parental and cisplatin-resistant BCa cell lines by diminishing tumor cell adhesion and motility in vitro. Hence, ART could be beneficial to supplement the therapy of advanced BCa patients. Further investigations, also in vivo, are necessary for verification.

## 5. Conclusions

ART inhibited metastatic potential in both parental and cisplatin-resistant BCa cells in a cell type-dependent manner. Specifically, ART diminished the adhesive capacity and motility of parental and cisplatin-resistant BCa cells. Inhibition of the metastatic potential of the BCa cells by ART was accompanied by distinct modulations of integrin subtypes (surface and total) in a cell type-dependent manner. The functional relevance of these alterations has been shown in corresponding blocking studies. Thus, ART might be a promising treatment option for patients with advanced or therapy-resistant BCa.

## Figures and Tables

**Figure 1 cells-14-00570-f001:**
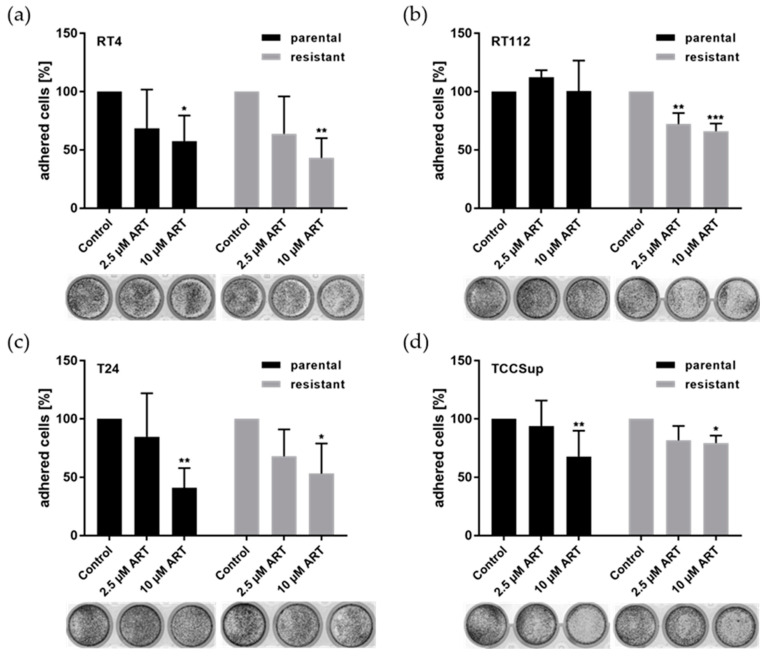
Tumor cell adhesion to collagen. Adhesion of parental and cisplatin-resistant BCa cells, RT4 (**a**), RT112 (**b**), T24 (**c**), and TCCSup (**d**) to collagen after exposure to ART. Tumor cells were pre-treated with ART [2.5 or 10 µM] for 48 (RT112, T24, and TCCSup cells) or 72 h (RT4 cells). Adhesion was evaluated after 30 min incubation. Untreated cells served as controls, set to 100%. Pictures under bars represent adhered BCa cells to collagen. Error bars indicate standard deviation (SD). Significant difference to untreated control: * = *p* ≤ 0.05, ** = *p* ≤ 0.01, *** = *p* ≤ 0.001. *n* = 4.

**Figure 2 cells-14-00570-f002:**
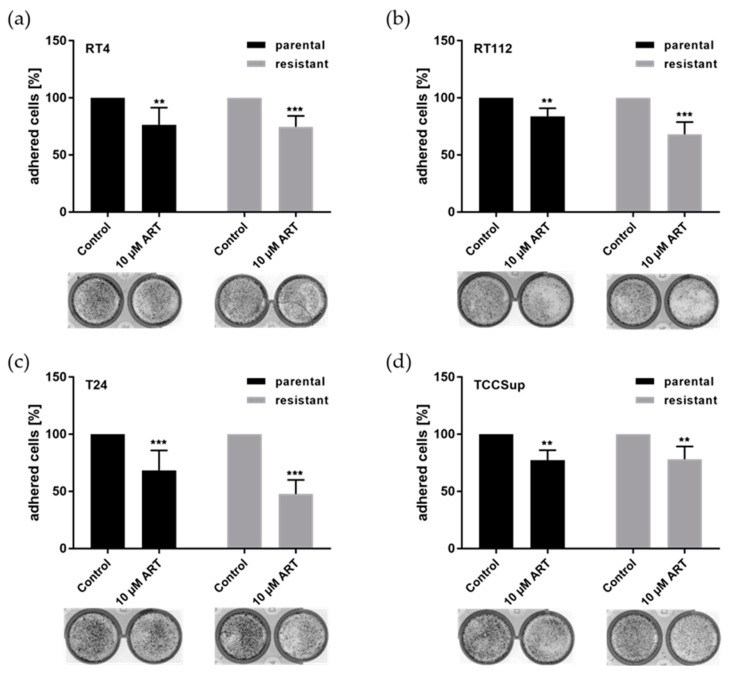
Tumor cell adhesion to vascular endothelial cells. Adhesion of parental and cisplatin-resistant BCa cells, RT4 (**a**), RT112 (**b**), T24 (**c**), and TCCSup (**d**) to HUVECs after treatment with ART [10 µM] for 48 (RT112, T24, and TCCSup cells) or 72 h (RT4 cells). Adhesion was investigated 2 h after incubation. Untreated cells served as controls, set to 100%. Pictures under the bars represent adhered tumor cells. Error bars indicate standard deviation (SD). Significant difference to untreated control: ** = *p* ≤ 0.01, *** = *p* ≤ 0.001. *n* = 4.

**Figure 3 cells-14-00570-f003:**
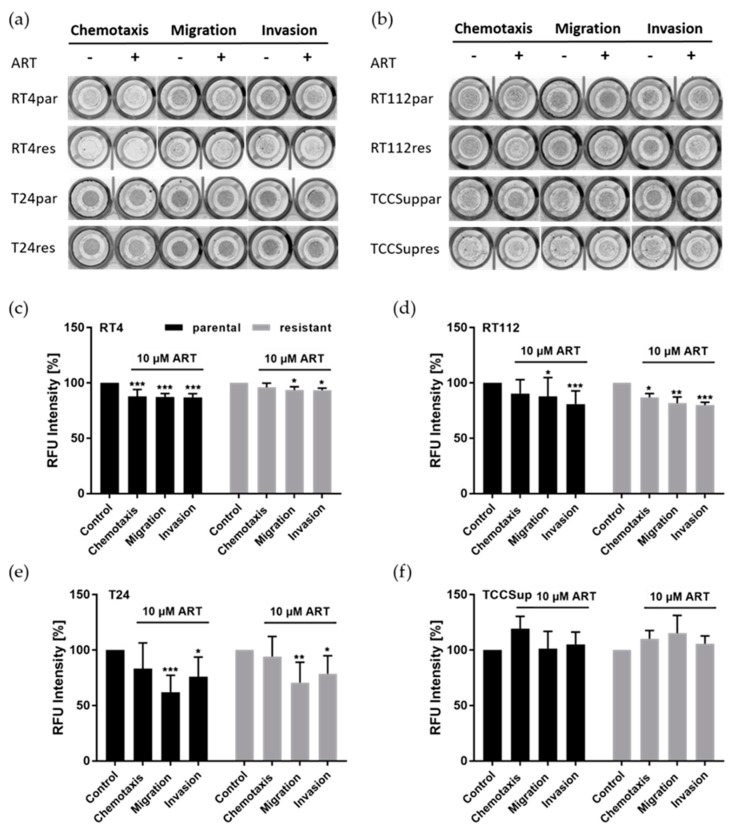
Tumor cell chemotaxis, migration, and invasion after ART application. Representative chemotaxis, migration, and invasion pictures of parental (par) and cisplatin-resistant (res) RT4 (**a**), RT112 (**a**), T24 (**b**), and TCCSup (**b**) cells after treatment with ART. Analysis of chemotaxis, migration, and invasion of RT4 (**c**), RT112 (**d**), T24 (**e**), and TCCSup (**f**) cell lines pre-treated with ART [10 µM] for 48 (RT112, T24, and TCCSup cells) or 72 h (RT4 cells) evaluated after 24 h incubation in the transwell system, compared to untreated controls. Relative fluorescence (RFU) of untreated controls was set to 100%. Error bars indicate standard deviation (SD). Significant difference to untreated control: * = *p* ≤ 0.05, ** = *p* ≤ 0.01, *** = *p* ≤ 0.001. *n* = 5.

**Figure 6 cells-14-00570-f006:**
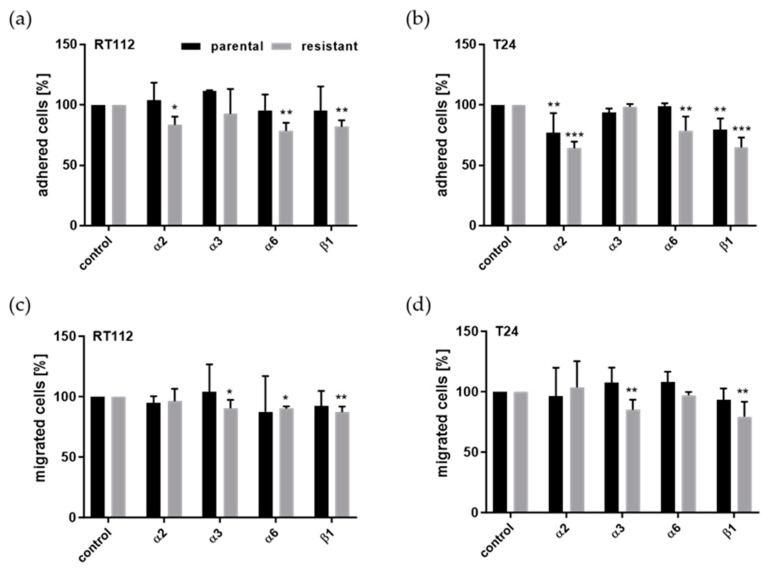
Validation of the functional relevance of ART-altered integrin-subtypes in the BCa cells. Cell adhesion and cell invasion of RT112 (**a**,**c**) and T24 (**b**,**d**) cells after blockade of integrin α2, α3, α6, or β1. Percentage of bound or invaded cells, compared to the untreated controls. The relative fluorescence (RFU) of untreated controls was set to 100%. Error bars indicate standard deviation (SD). Significant difference to untreated control: * = *p* ≤ 0.05, ** = *p* ≤ 0.01, *** = *p* ≤ 0.001. *n* = 4.

## Data Availability

The raw data supporting the conclusions of this article will be made available by the authors on request.

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
