# Peer review of "Artesunate Inhibits Metastatic Potential in Cisplatin-Resistant Bladder Cancer Cells by Altering Integrins"

_cells, 2025, doi:10.3390/cells14080570_

Round 1

Reviewer 1 Report

Comments and Suggestions for Authors

Authors have previously published on this topic: they discussed cell growth, proliferation, and cell cycle phases were investigated, as were apoptosis, necrosis, ferroptosis, autophagy, metabolic activity, and protein expression, but not metastasis. Abstract needs to be modified to state this difference between the previous publication and the current manuscript.

They show that ART inhibits cell adhesion, chemotaxis, migration and invasion, and suppressed the cell surface expression of integrins. However, it is not clear how ART causes integrin blockade, and whether these effects alter metastasis in vivo.

Other points to consider.

  1. Explain in the abstract what is artesunate (use language from lines 65-69), and why it is being used as therapy.
  2. RT4 cells are naturally cisplatin resistant. So, to some extent, are T24 cells. Therefore, it is not clear how much difference artificially making these cells resistant to cisplatin will make.
  3. Line 49: This manuscript is from Germany and UK, but quote statistics from the US. Are similar numbers available from Western Europe?
  4. Innumerable grammatical errors.
Comments on the Quality of English Language

Examples of grammar that can be improved: 

Line 47: "have shown to relapse" change to "have been shown to relapse"

Please re-read your manuscript prior to resubmission. 

Reviewer 2 Report

Comments and Suggestions for Authors

Please see attached file below 

Round 2

Reviewer 1 Report

Comments and Suggestions for Authors

The authors have satisfactorily responded to all previous criticisms. 

Comments on the Quality of English Language

The grammar has improved significantly, but there are still minor errors that can be improved. Please read the manuscript carefully to ensure grammatical errors are eliminated. 

Reviewer 2 Report

Comments and Suggestions for Authors

The authors have revvised their manuscript thoughtfully and thourougly. This reviewer did not identify any obious weekness. Overall thisis  much improved submission.